**Data Availability Statement:** Raw sequence data are available in NCBI GEO accession GSE199253 and GSE163160 (https://www.ncbi.nlm.nih.gov/

# Mapping genetic effects on cell type-specific chromatin accessibility and annotating complex immune trait variants using single nucleus ATAC-seq in peripheral blood

Paola Benaglio[1◉], Jacklyn Newsome[2◉], Jee Yun Han[3], Joshua Chiou[4], Anthony Aylward[2], Sierra Corban[1], Michael Miller[3], Mei-Lin Okino[1], Jaspreet Kaur[1], Sebastian Preissl[3], David U. Gorkin[3], Kyle J. Gaulton[1]*

**1** Department of Pediatrics, University of California San Diego, San Diego, California, United States of America, **2** Bioinformatics and Systems Biology Program, University of California San Diego, San Diego, California, United States of America, **3** Center for Epigenomics, Department of Cellular and Molecular Medicine, University of California San Diego, San Diego, California, United States of America, **4** Biomedical Sciences Graduate Program. University of California San Diego, San Diego, California, United States of America

◉ These authors contributed equally to this work.

* kgaulton@ucsd.edu

## Abstract

Gene regulation is highly cell type-specific and understanding the function of non-coding genetic variants associated with complex traits requires molecular phenotyping at cell type resolution. In this study we performed single nucleus ATAC-seq (snATAC-seq) and genotyping in peripheral blood mononuclear cells from 13 individuals. Clustering chromatin accessibility profiles of 96,002 total nuclei identified 17 immune cell types and sub-types. We mapped chromatin accessibility QTLs (caQTLs) in each immune cell type and sub-type using individuals of European ancestry which identified 6,901 caQTLs at FDR < .10 and 4,220 caQTLs at FDR < .05, including those obscured from assays of bulk tissue such as with divergent effects on different cell types. For 3,941 caQTLs we further annotated putative target genes of variant activity using single cell co-accessibility, and caQTL variants were significantly correlated with the accessibility level of linked gene promoters. We fine-mapped loci associated with 16 complex immune traits and identified immune cell caQTLs at 622 candidate causal variants, including those with cell type-specific effects. At the 6q15 locus associated with type 1 diabetes, in line with previous reports, variant rs72928038 was a naïve CD4+ T cell caQTL linked to *BACH2* and we validated the allelic effects of this variant on regulatory activity in Jurkat T cells. These results highlight the utility of snATAC-seq for mapping genetic effects on accessible chromatin in specific cell types.

geo/query/acc.cgi?acc=GSE199253; https://www.
ncbi.nlm.nih.gov/geo/query/acc.cgi?acc=
GSE163160). Genotyping data can be accessed
upon request through the European Genome-
Phenome Archive (EGA) at ID EGAS00001006184
(https://ega-archive.org/datasets/
EGAD00010002308). All processed data
underlying analyses and figures in the paper are
located at https://doi.org/10.5281/zenodo.7375095
and in the supplementary tables where specified.
Pipelines and custom code for all analyses and
graphs of this manuscript are available in the
Github page https://github.com/Gaulton-Lab/
pbmc_snATAC.

**Funding:** This work was supported by National
Institute of Diabetes and Digestive and Kidney
Diseases awards DK112155, DK120429 and
DK122607 to K.J.G. The funders had no role in
study design, data collection and analysis, decision
to publish, or preparation of the manuscript.

**Competing interests:** I have read the journal's
policy and the authors of this manuscript have the
following competing interests: Dr. Gaulton has
done consulting for Genentech and holds stock in
Neurocrine Biosciences. Dr. Benaglio is an
employee of Shoreline Bioscience. Dr. Chiou is an
employee and shareholder of Pfizer. These
affiliations have no competing interest related to
the submitted work. The other authors have no
competing interests to disclose. The funders had
no role in study design, data collection and
analysis, decision to publish, or preparation of the
manuscript.

## Author summary

In this study we profiled regulatory elements in specific immune cell types and sub-types
in peripheral blood using single cell experiments in 13 individuals. We then identified
genetic variation between individuals associated with the activity of regulatory elements
in each cell type and sub-type. We finally used these results to identify genetic variants
associated with blood traits and autoimmune disease risk that alter immune cell type regu-
lation. In one example, we show that a type 1 diabetes risk variant alters T cell regulatory
element activity near the *BACH2* gene, suggesting a possible mechanism for how this vari-
ant may affect disease. Together these findings reveal genetic variants which alter the
activity of regulatory elements in specific immune cell types and sub-types including
those that are involved in common traits and disease risk.

## Introduction

Genome-wide association studies have identified thousands of genomic loci associated with
complex human traits and disease [1–3], but their molecular mechanisms remain largely
unknown. Interpreting the mechanisms of trait-associated loci is paramount to an improved
understanding of the cell types, genes and pathways involved in complex traits and disease [1].
Genetic variants at complex trait-associated loci are primarily non-coding and enriched in
transcriptional regulatory elements [1,4,5], implying that the majority affect gene regulatory
programs. As gene regulation is highly cell type-specific [6,7], uncovering the molecular mech-
anisms of complex trait loci requires determining the function of non-coding variants in the
individual cell types that comprise a tissue. While substantial advances have been made in
annotating the non-coding genome [5,8], the regulatory effects of genetic variants in specific
cell types are still largely unknown.

Mapping quantitative trait loci (QTLs) for molecular phenotypes such as gene expression,
histone modifications and chromatin accessibility is an effective strategy to determine the reg-
ulatory activity of genetic variants [9–15]. Molecular QTL studies to date have been primarily
performed in "bulk" tissue, cell lines, or individual sorted cell types, however, and therefore
have not yet widely annotated the breadth of cell type effects. Single cell technologies have cre-
ated new avenues to study gene regulation in the specific cell types comprising a heterogeneous
tissue and define relationships to complex traits and disease [16,17]. Several recent studies
have mapped QTLs using cell type-specific profiles derived from single cell assays [18–20].
These studies represented proof-of-concept for using profiles derived from single cell data to
map genetic effects on molecular phenotypes in specific cell types and sub-types. Moreover,
they enabled additional analyses which leveraged data from across thousands of cells such as
the identification of co-expression QTLs [19].

In this study we used single nucleus ATAC-seq (snATAC-seq) to profile human periph-
eral blood mononuclear cell (PBMC) samples. We derived chromatin accessibility profiles of
immune cell types and sub-types and mapped chromatin accessibility QTLs (caQTLs) for
these profiles which identified thousands of immune cell type and sub-type caQTLs. We
characterized caQTLs for each cell type, including caQTLs whose effects are obscured in
"bulk" assays, and linked distal caQTLs to putative target gene promoters using single cell
co-accessibility. Finally, we fine-mapped causal variants at genomic loci associated with 16
complex immune traits and diseases, annotated fine-mapped variants for these traits with
immune cell type caQTLs and validated the molecular effects of high-probability caQTL
variants.

## Results

### Chromatin accessibility profiling of peripheral blood mononuclear cells

We performed snATAC-seq and genotyping of human peripheral blood cell (PBMC) samples to map genetic effects on lymphoid and myeloid cell type accessible chromatin (Fig 1A). We used droplet-based snATAC-seq (10X Genomics) to assay 13 PBMC samples from individuals of self-reported European descent (S1 Table, see Methods). The snATAC-seq libraries were sequenced to an average depth of 212M read pairs (24,227 read pairs per nucleus on average), and libraries had consistently high-quality metrics including enrichment at transcription start sites (TSS) and fraction of reads mapping in peaks (S2 Table). We then performed array genotyping of each sample and imputed genotypes into 308M variants in the TOPMed r2 reference panel. Principal components analysis of genotypes mapped onto 1000 Genomes Project data confirmed European ancestry for 10 out of 13 samples (S1 Fig).

After extensive quality control that removed low quality cells and potential doublet cells (see Methods), we performed clustering of 96,002 snATAC-seq profiles, which revealed 17 clusters (Fig 1B). We assigned clusters to lymphoid and myeloid cell types and sub-types based on the chromatin accessibility patterns at known marker genes (Fig 1B, S1, S2 and S3 Tables).

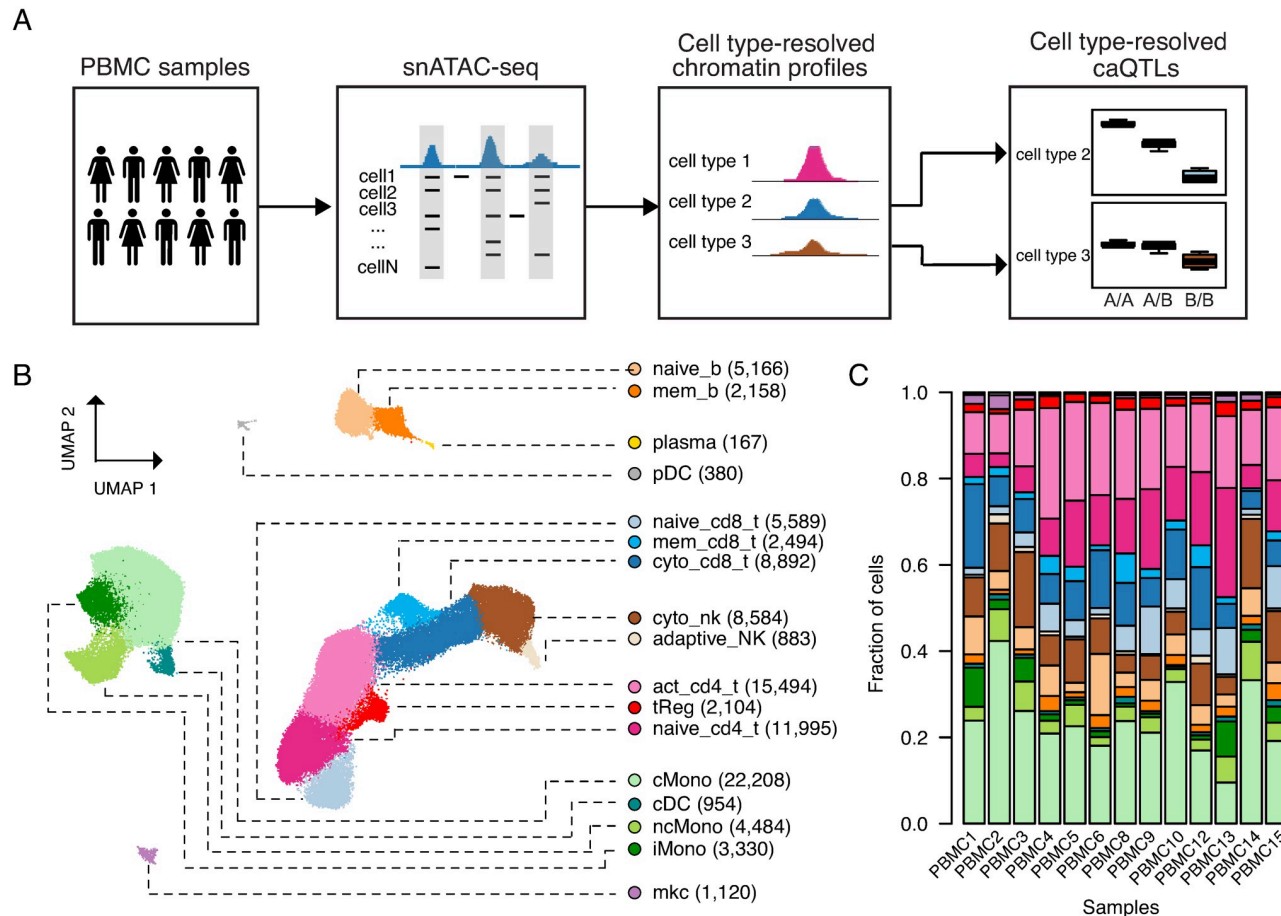

**Fig 1. Single nucleus ATAC-seq in a population of PBMC samples.** A) Schematic overview of the study. B) Clustering of single cell accessible chromatin profiles of 96,002 PBMCs from 13 individuals. Cells are plotted based on the first two UMAP components. Seventeen distinct clusters, indicated by different colors, were identified and assigned to a cell type based on known marker genes. The number of cells for each cell type is indicated in parenthesis. C) Barplot showing the relative proportions of each cell type in each sample. Color scheme is the same as in 1B.

For example, among major immune cell types, *NCR1* accessibility marked NK cells and *MS4A1* accessibility marked B cells. Among cell sub-types, accessibility at *FOXP3* differentiated regulatory T cells from other T cell sub-types, and accessibility at *TCL1A* differentiated naïve B cells from memory B cells (S2A and S3 Figs). To confirm the cell type and sub-type identity of snATAC-seq clusters, we compared genes specific to each cluster to those derived from published single cell RNA-seq (scRNA-seq) data in PBMCs [21] and in each case identified strongest correlation with the corresponding cell type or sub-type (S2C Fig). The proportion of each immune cell type and sub-type was broadly consistent across samples (Figs 1C and S4A) and was highly correlated with broad cell proportions determined from flow cytometry of cell surface markers for each sample (S4 Table and S4B and S4C Fig). Similarly, clusters were composed of similar proportions of cells from different individuals (S4D Fig). These results demonstrate that snATAC-seq of PBMCs resolved lymphoid and myeloid cell types and sub-types with broadly consistent representation across samples.

## Mapping chromatin accessibility QTLs in immune cell types and sub-types

Within each immune cell type and sub-type cluster, we aggregated reads for all cells in the cluster, generated accessible chromatin read count profiles, and called accessible chromatin sites using MACS2 [22]. Considering all immune cell types and sub-types there were 243,247 total accessible chromatin sites (S5 Table). Immune cell type and sub-type sites were highly concordant with sites identified in a previous study of FACS-sorted immune cell types [23] (S5 Fig). For each site, we first removed reads mapping to only one allele at polymorphic bases using WASP [24] and then performed QTL mapping of chromatin accessibility read counts using RASQUAL [25], a method which combines population-based and allele-specific mapping. We mapped QTLs at both the cell type (Monocytes, B, T and NK cells) and sub-type (all 17 clusters) resolution. For each cell type or sub-type, we retained sites with >5 reads per sample on average and only tested variants that mapped directly in accessible sites and were heterozygous in at least two samples. After applying these filters, on average 80,029 variants per cell type were tested for association with 49,396 peaks (4.5 variants/peak). For comparison, we also performed caQTL mapping after merging all reads for each sample ignoring their cell of origin to mimic a "bulk" ATAC-seq experiment.

In total we identified 6,901 distinct caQTLs in an immune cell type or sub-type (FDR < .10), including 5,908 at cell type resolution (666 to 3,507 in each cell type) and 5,272 at sub-type resolution (0 to 2,941 in each cell sub-type) (Fig 2A and S6 Table). A consistent proportion of caQTLs reached more stringent FDR thresholds (S6A Fig). For example, at FDR < .05 we identified 4,220 caQTLs, including 3,162 at cell type resolution and 3,654 at sub-type resolution. We evaluated properties of cell type and sub-type caQTLs, which revealed that the overall distribution of caQTL associations appeared well-calibrated (S6B Fig), and only a small proportion of caQTLs (0.6%) failed allele mapping bias ($\psi < 0.2$, $> 0.8$) and sequencing mapping error rate ($\partial > 0.1$) metrics (S6 Table). In addition, the caQTLs had on average higher coverage than other tested variants and 94.8% had >10 avg. reads per sample (S7 Fig), and caQTLs were enriched for variants where all three genotypes (hom. ref, het., hom. alt) were observed compared to only two genotypes (S8A–S8D Fig). We also evaluated the impact of potential misclassified or residual doublet cells in each cluster on cell type-specific caQTL discovery. For each cell type cluster, we removed cells with accessibility at the promoter of several genes highly specific to other cell types and repeated caQTL mapping (see Methods). After removing these cells, the effect sizes of caQTLs for each cell type were almost perfectly correlated (>.99) and directionally concordant (>99.8%), suggesting limited impact on the caQTL results.

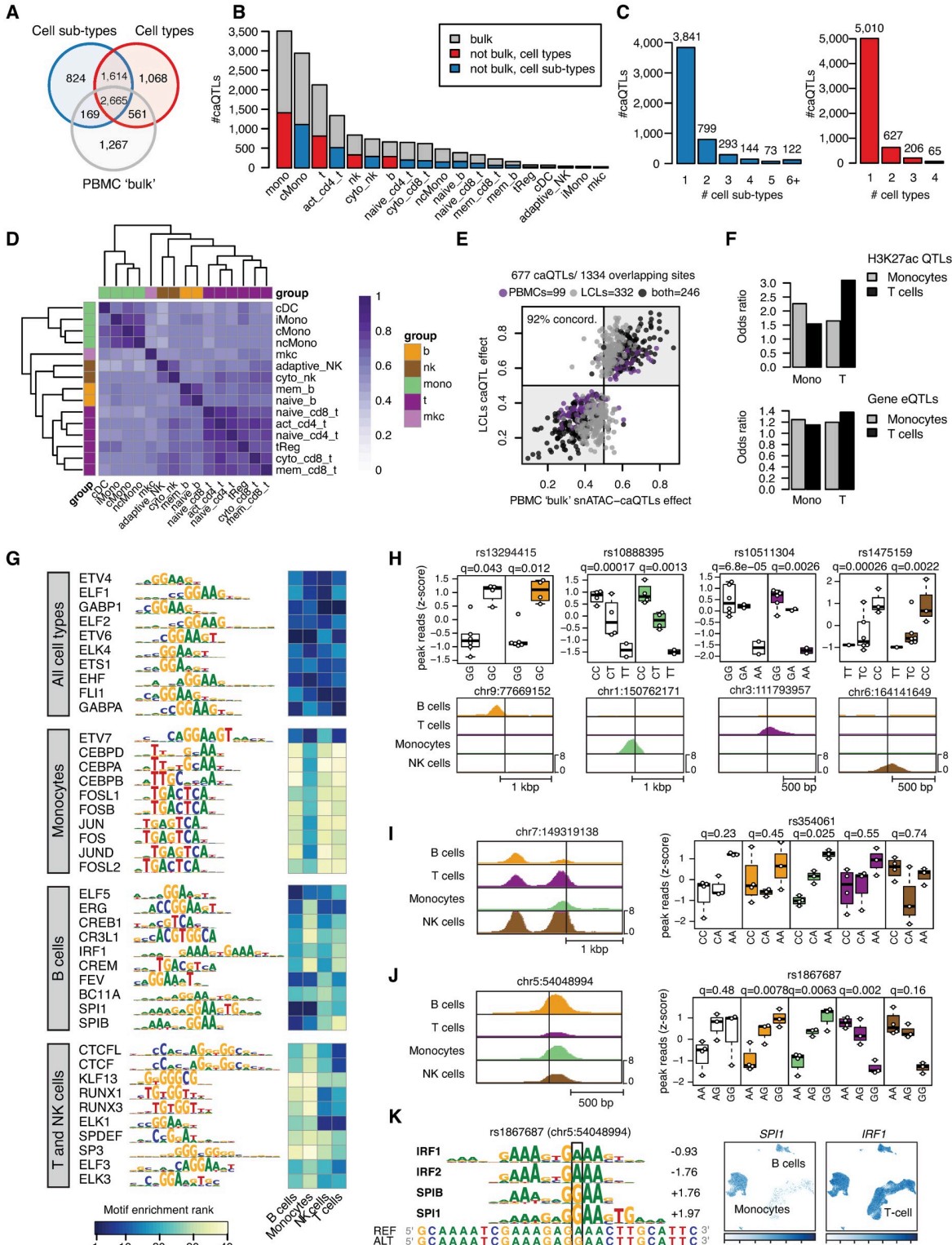

**Fig 2. Identification and characterization of immune cell type chromatin accessibility QTLs.** A) Venn diagram showing the total number of caQTLs in immune cell types (red), cell sub types (blue); and "bulk" (gray). B) Number of caQTLs identified in each cell type and the subset found in "bulk" data. C) Number of caQTLs unique or common to different cell types. D) Heatmap of pairwise Spearman correlation between effect sizes of caQTLs identified in at least one of the two cell types in the pair. E) Scatter plot of effect sizes for PBMC "bulk" caQTLs from this study or published caQTLs from 24 LCLs. F) Overlap between cell type caQTLs and H3K27ac QTLs (top) or

expression QTLs (bottom) in either monocytes (gray) or T cells (black). G) Transcription factor motifs disrupted by caQTLs across different cell types. The heatmap shows the enrichment ranking of TFs in each cell type. H) Examples of caQTLs in peaks specific to a single cell type including rs13294415, rs10888395, rs10511304 and rs1475159. Top panels: colored-coded boxplots show association in different cell types and white boxplots show "bulk" PBMCs. Association q-values are shown on the top and variant genomic location (hg19) is shown at the bottom. Bottom panels: genome browser view of cell type chromatin profiles. I) Variant rs354061 mapped in a peak active in all cell types but was a caQTL in monocytes only. Left: genome browser view of cell type chromatin signal. Right: boxplots as in H. J) Variant rs1867687 was a significant caQTL in all cell types but had opposite effects in different cell types. Left: genome browser view of cell type chromatin signal. Right: boxplot of signal by genotype in each cell type and "bulk". Boxplots are color-coded as in I. K) Left: TF motifs altered by rs1867687 and their respective score differences. Positive scores indicate preference for alternate allele. Right: UMAP plot showing accessibility of the *SPI1* and *IRF1* genes.

Given the large number of caQTLs identified relative to our sample size, we next investigated parameters driving caQTL discovery. As previous studies have mapped caQTLs in bulk tissue, we first determined the impact of having tested each cell type individually. When treating each sample as "bulk" there were 4,662 caQTLs (Fig 2A and S6 Table), a 32% reduction in the caQTLs identified when separated by cell type. Among caQTLs for each cell type also identified in "bulk", there were also consistently larger effects at the cell type level (S6D Fig). Next, we determined the impact of sequencing depth by down-sampling data to 25% (53M read-pairs on average), which identified 2,503 caQTLs in "bulk", a 46% reduction compared to using full sequence data. We further determined the contribution of allelic imbalance to caQTL mapping. Excluding the allelic imbalance component entirely resulted in substantially fewer caQTLs (535 total across cell types, 699 across sub-types), although the allelic effects were highly concordant with the full caQTL results (S9 Fig). These results demonstrate that the cell type resolution, sequencing depth, and allelic imbalance are all contributors to the number of caQTLs identified.

The majority of caQTLs were identified at FDR < .10 at multiple resolutions ("bulk", cell type, sub-type) (Fig 2A). The number of caQTLs identified in each cell type was proportional to the number of cells observed for that cell type (Figs 2B and S6C), likely due to differences in read depth leading to reduced power for less common cell types. To further determine the impact of cell number on caQTL discovery, we down-sampled cells from the most common cell type classical monocytes and re-performed caQTL mapping (see Methods). As expected, the number of detected caQTLs decreased as a function of the number of cells. Below 100 cells per sample, almost no caQTLs were detected at FDR < .10 and the proportion of caQTLs estimated to be shared ($\pi1$) with the original caQTLs decreased substantially (S7 Table). Most caQTLs were identified at FDR < .10 in only one cell type or sub-type (85% for cell types, 73% for cell sub-types) (Fig 2C). The caQTLs identified at FDR < .10 in only one cell type had comparable quality metrics to those identified in multiple cell types, although the average read depth was lower (S10A Fig). However, when considering caQTLs significant in at least one cell type, the allelic effects ($\pi$) were highly correlated across cell types (median Spearman correlation $r = 0.62$, Fig 2D), with stronger correlation between more similar cell types. Furthermore, only a small proportion of caQTLs had significant evidence for heterogeneity (ANOVA FDR < .10) in allelic effects across cell types and sub-types (122 and 339, respectively) (S10A Fig). Finally, caQTLs in each cell type and sub-type showed a high degree of sharing with "bulk" caQTLs (median $\pi1 = 0.96$) (S7 Table). Together these results support that many caQTLs are likely shared across cell types.

We next compared cell type caQTLs in our study to external QTL datasets previously generated in immune cells. We first compared caQTL results from our "bulk" analysis with caQTLs mapped in 24 lymphoblastoid cell lines (LCLs) [25]. Of the 1,942 caQTLs in our data that were tested in the LCL study, 541 (28%) were also significant LCL caQTLs (OR = 15.7, P<2.2x10^-16, Fisher's exact test). Furthermore, among 677 caQTLs identified in either study where the same

variants were tested in both studies, there was 92% concordance in the direction of effect (Fig 2E). Of note, when considering caQTLs from each individual cell type, B cell caQTLs had the highest overlap with LCL caQTLs, consistent with LCLs being derived from B cells (S11 Fig). We next compared caQTLs in our study to published histone H3K27ac QTLs (hQTLs) and expression QTLs (eQTLs) from FACS-sorted T cells and Monocytes from the BLUEPRINT project [15]. The enrichment for T cell hQTLs was stronger in T cell caQTLs (OR = 3.1, $P = 6 \times 10^{-117}$) compared to monocyte caQTLs (OR = 1.5, $P = 4.4 \times 10^{-15}$) (Fig 2F). We observed the converse pattern for monocyte hQTLs, which were more enriched for monocyte caQTLs than T cell caQTLs (T cell OR = 1.65, $P = 2.1 \times 10^{-22}$; monocyte OR = 2.3, $P = 1.9 \times 10^{-119}$) (Fig 2F). We observed the same cell type enrichment pattern for T cell and monocyte eQTLs (T-cell eQTLs: T-cell OR = 1.37, $P = 6.1 \times 10^{-28}$; monocyte OR = 1.15, $P = 7.5 \times 10^{-08}$; monocyte eQTLs: T cell OR = 1.19, $P = 9.3 \times 10^{-10}$; monocyte OR = 1.24, $P = 1.6 \times 10^{-120}$; Fig 2F), confirming the reproducibility of our cell type caQTLs with independent cell type-specific QTLs.

To identify transcription factors (TFs) mediating immune cell type caQTLs, we identified TF sequence motifs preferentially disrupted by caQTL variants in each cell type. We used MotifBreakR [26] to predicted allelic effects of SNPs on TF motifs from the HOCOMOCO v10 human database [27], comprising 640 motifs corresponding to 595 unique TFs. We first predicted allelic motif effects for all variants tested for QTL association. Then, for each TF motif, we compared the proportion of motif instances disrupted by caQTLs compared to non-caQTL variants. Thus, we were able to measure the enrichment of predicted TF-disrupting caQTLs for each TF motif. Immune cell type caQTLs were broadly enriched for disrupting any TF motif compared to non-caQTL variants (OR = 1.24, $P = 1.1 \times 10^{-4}$, Fisher's exact test). When considering caQTLs in each cell type, there were 29 TF motifs significantly enriched for B cell caQTLs, 39 motifs enriched for T cell caQTLs, 35 motifs enriched for NK cell QTLs and 86 motifs enriched for monocyte QTLs (FDR<0.05, one-tailed binomial test, Fig 2G and S8 Table). Motifs disrupted by caQTLs included those with broadly shared enrichment across different cell types such as ETS family motifs, as well as those with cell type-specific enrichment such as FOS/JUN and CEBP motifs in monocytes, and RUNX motifs in NK and T cells. As TFs in the same structural sub-family often share similar binding motifs, we next determined the expression of TF genes using published PBMC scRNA-seq. We identified TF genes with cell type-specific expression (FDR < .05, LogFC>1; Wilcoxon test) that matched motif sub-family enrichment patterns (S8 Table), implicating specific TFs likely altered by immune cell type caQTLs.

At numerous loci, caQTLs mapped at cell type and sub-type resolution provided insight beyond those obtained by mapping caQTLs in "bulk" tissue. The most straightforward examples consisted of caQTLs for accessible chromatin sites active in only one cell type, where the effects of a caQTL identified in "bulk" data could be simply ascribed to that cell type (1,947 caQTLs, S10B Fig). For example, rs13294415 was a caQTL for a B cell-specific site (allelic effects [π] = .73, q-value = .012), rs10888395 was a caQTL for a monocyte-specific site (π = .34, q = .0013), rs10511304 was a caQTL for a T cell-specific site (π = .26, q = .0026) and rs1475159 was a caQTL for a NK cell-specific site (π = .77, q = .0022) (Fig 2H). We also identified caQTLs for immune sub-type-specific sites (883 caQTLs), such as rs1957554 which was a caQTL for a naïve B-cell-specific site (π = .32, q = .038) and rs855166 which was a caQTL for a naïve CD4 + T cell-specific site (π = .77, q = .0021) (S12A Fig). Another class of caQTLs were those for sites active in all cell types, yet where the variant effects were restricted to one or several cell types. For example, variant rs354061 mapped in a site active in all immune cell types and had a significant effect in monocytes (π = .67, q = .023) but no effect in other cell types (ANOVA q = .0014) (Fig 2I). In this example, variant effects in "bulk" data were dampened due to the inclusion of cell types with no effect (rs354061 π = .54, q = .98) (Figs 2I and S12B).

We also observed caQTLs with more complex effects, such as those with divergent effects on different cell types. For example, variant rs1867687 was a significant caQTL in all immune cell types yet had opposed effects across cell types (ANOVA q = 4.4x10$^{-4}$), where the G allele had increased accessibility in B cells and monocytes (B cell $\pi$ = .66, q-value = .008; monocyte $\pi$ = .62, q = .0063) and the A allele had increased accessibility in T cells and NK cells (T $\pi$ = .25, q = .002; NK cell $\pi$ = .31, q = .16) (Fig 2J). In comparison, rs1867687 had no effect in "bulk" data ($\pi$ = .53, q = .48). The alleles of this variant were predicted to bind different TFs, where the G was predicted to bind SPI1 and SPIB motifs and the A allele was predicted to bind IRF TF motifs (Fig 2K). SPI1 and SPIB motifs were specifically enriched in B cells and monocytes, whereas IRF motifs were broadly enriched across cell types (Fig 2K), suggesting a potential mechanism through which this variant has opposing effects on different immune cell types.

## Linking distal caQTLs to effects on target gene promoters

Among the 6,901 immune cell type and sub-type caQTLs identified in our study, a minority (19%) mapped to gene promoter regions. The remaining caQTLs were in chromatin sites distal to promoters, and we therefore sought to define the target genes of these caQTLs. Co-accessibility between pairs of accessible chromatin sites across single cells has been used to annotate putative target genes of distal enhancers [17,28]. We therefore defined co-accessible sites (co-accessibility >.05) in the major cell types (monocytes, T, B, and NK cells) using Cicero, where the threshold for co-accessibility was determined based on enrichment for 3D chromatin interactions (S13A Fig) as well as based on previous studies [28]. For each cell type we retained co-accessible sites greater than 10kb apart and that also were co-accessible in at least four samples individually. In total we identified 770,015 pairs of co-accessible sites, which included between 152k and 327k per cell type (Fig 3A). We compared co-accessible sites for each cell type to 3D chromatin interactions from promoter capture Hi-C (pCHi-C) data previously generated in 16 immune cell types and sub-types [29]. We observed strongest enrichment of cell type co-accessible sites for the corresponding cell type in pCHi-C interactions in each case, except for NK cells, which were not assayed by pCHi-C (Fig 3B). When segregating co-accessible sites by distance bins, we observed enrichment for pCHi-C loops across all bins with stronger enrichment at greater distances (Figs 3C and S13B).

Using the co-accessible sites identified for each cell type we then annotated caQTLs with their putative target genes. There were 254,771 distal accessible chromatin sites co-accessible with at least one promoter site (57k-103k per cell type) and 84,788 promoter sites co-accessible with promoter sites of a different gene (25k-37k per cell type) (Fig 3D and 3E). Across all 6,901 caQTLs, 3,941 were either in a site co-accessible with at least one gene promoter or in a promoter site directly. Among these 3,941 caQTLs, the majority were distal sites co-accessible with a promoter (61–64% per cell type) (Fig 3F). Among distal caQTLs co-accessible with a gene promoter, 34–47% were linked to just one gene (Fig 3G).

Previous studies have identified coordinated allelic effects between distal sites and interacting promoters [30]. We therefore tested caQTL variants for association with chromatin accessibility levels of all promoter sites co-accessible with the caQTL site. There was a positive and highly significant correlation between variant allelic effects on the original site and effects on co-accessible promoter sites (B r = .24, T r = .27, monocyte r = .23, NK r = .27, Pearson correlation) (Fig 3H). When separating co-accessible sites by distance, the correlations were reduced between more distal sites (S13C Fig). As we were unable to leverage allelic imbalance in this analysis, our power was more limited, and we only identified 19 linked promoter caQTLs at FDR < .20. For example, at the 14q32 locus rs12434305 was a caQTL for a distal site in T cells ($\pi$ = .31, q = .0026) and was also a caQTL for the *GSC* promoter linked to the distal site ($\pi$ =

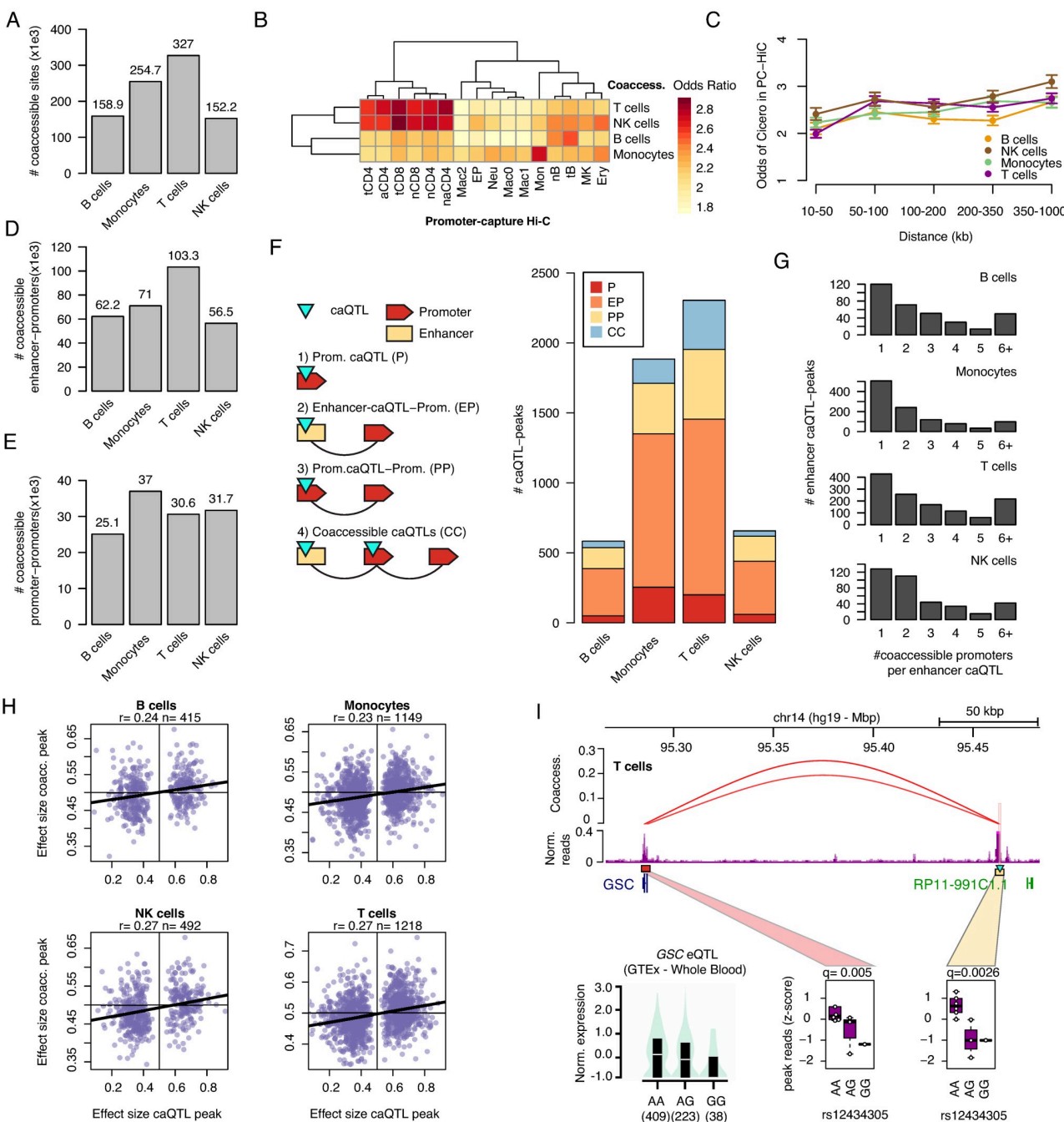

**Fig 3. Linking distal immune cell caQTLs to putative target genes.** A) Number of co-accessible links in each immune cell type. B) Enrichment of cell type co-accessible links for overlap with promoter-capture Hi-C (pcHi-C) interactions in immune cell types. C) Enrichment of cell type co-accessible links for pcHi-C interactions separated by distance between linked sites. D) Number of co-accessible links between a promoter site (+/- 1kb) and a distal (non-promoter) site in each cell type. E) Number of co-accessible links between promoter sites. F) Breakdown of caQTLs linked to promoters in each cell type, including caQTLs directly in promoter sites, caQTLs in distal sites co-accessible with a promoter site, caQTLs in promoter sites co-accessible with a different promoter site, and more complex cases involving multiple linked caQTLs. G) Breakdown of caQTLs in each cell type by the number of promoter sites they were linked to. H) Correlation in the effects of caQTL variants on the primary site and co-accessible promoter sites in each cell type. Pearson correlation coefficient and number of co-accessible pairs of peaks are indicated. I) Variant rs12434305 was a caQTL for a T cell site as well as a caQTL for a co-accessible site at the *GSC* promoter and a QTL for *GSC* expression in GTEx.

.34, q = .005) (Fig 3I). This variant was also a QTL for the expression of *GSC* in whole blood in GTEx v8 [9] (NES = -.22, P = 1.8x10$^{-5}$), and which was directionally consistent with the C allele having increased activity. Together these results demonstrate how snATAC-seq data can link caQTLs to effects on putative target genes.

## Identifying caQTLs at fine-mapped variants for complex immune trait loci

Genomic loci affecting complex immune traits and disease are primary non-coding, and the causal variants and molecular mechanisms at these loci are largely unknown. We therefore used immune cell type and sub-type caQTLs to annotate variants associated with complex immune traits and disease. We first collected published genome-wide association summary statistics for 16 blood cell count, autoimmune, inflammatory and allergy traits imputed into reference panels with comprehensive variant coverage such as 1000 Genomes or the Haplotype Reference Consortium (Fig 4A and S9 Table). At most traits, fine-mapping of causal variant sets at associated loci was either not performed as part of the initial study or not made available. We therefore fine-mapped primary association signals at loci reported for these 16 traits using a Bayesian approach, from which we generated credible sets of variants representing 99% of the total posterior probability for each signal (see Methods). Across all traits there were 1,275 total credible sets, which contained a median of 16 variants, where traits with the smallest credible set sizes included monocyte count (median = 6.5 variants), basophil count (median = 7.5 variants) and rheumatoid arthritis (median = 9 variants). At 396 signals fine-mapping resolved credible sets to 5 or fewer variants (Fig 4A).

We next annotated fine-mapping credible sets with immune cell type and sub-type caQTLs. Of 4,925 fine-mapped variants in an immune cell type or sub-type peak, 74% (3,655) were tested for caQTL association. Binning fine-mapped variants in peaks by minor allele frequency (MAF), however, revealed that while most variants with MAF>.20 were tested for caQTL association a much smaller proportion were tested at lower frequencies (S14A Fig). In total, 622 credible set variants representing 221 association signals were immune cell type or sub-type caQTLs (Fig 4B). We determined whether fine-mapped variants for each trait were preferentially enriched for caQTLs from specific immune cell sub-types by comparing to a background of non-caQTL sites (see Methods). Most traits (12/16) showed nominal enrichment (P < .05) for caQTL peaks in at least one cell sub-type, several of which recapitulated known biology of cell types contributing to the trait (Fig 4C). For example, type 1 diabetes (T1D)-associated variants were enriched in T cell caQTLs (activated CD4+ T logOR = 1.62, p = 0.017; naive CD8+ T logOR = 2.1, p = 0.030), where T cells are the critical cell type in the pathogenesis of T1D [31]. Strong enrichments for other traits may similarly point to cell types involved in trait biology. For example, child onset asthma-associated variants were enriched in cytotoxic CD8+ T (logOR = 1.9, p = 0.028), memory CD8+ T (logOR = 2.4, p = 0.017) and memory B cell caQTLs (logOR = 3.3, p = 0.006) (Fig 4C), and atopic dermatitis-associated variants were strongly enriched in non-classical monocyte caQTLs (logOR = 4.3, p = 0.0001).

Among fine-mapped variants that were immune cell caQTLs, 374 had a posterior probability >1% and 235 were either in a distal site linked to a gene promoter or in a promoter site directly (Fig 4D and S10 Table). Among these, at multiple loci fine-mapped variant caQTLs replicated cell type-specific effects observed in previous studies [32]. For example, at the 5q11.2 locus associated with rheumatoid arthritis (RA), among the two candidate variants with highest causal probability rs28722705 (PPA = .70) and rs7731626 (PPA = 0.28) only rs7731626 mapped in an accessible chromatin site. This variant was a caQTL in T cells (π = 0.34, q = 0.003), and was co-accessible with the *IL6ST* and *ANKRD55* promoters (S10 Table). A previous study identified rs7731626 as likely causal for multiple sclerosis and RA and was

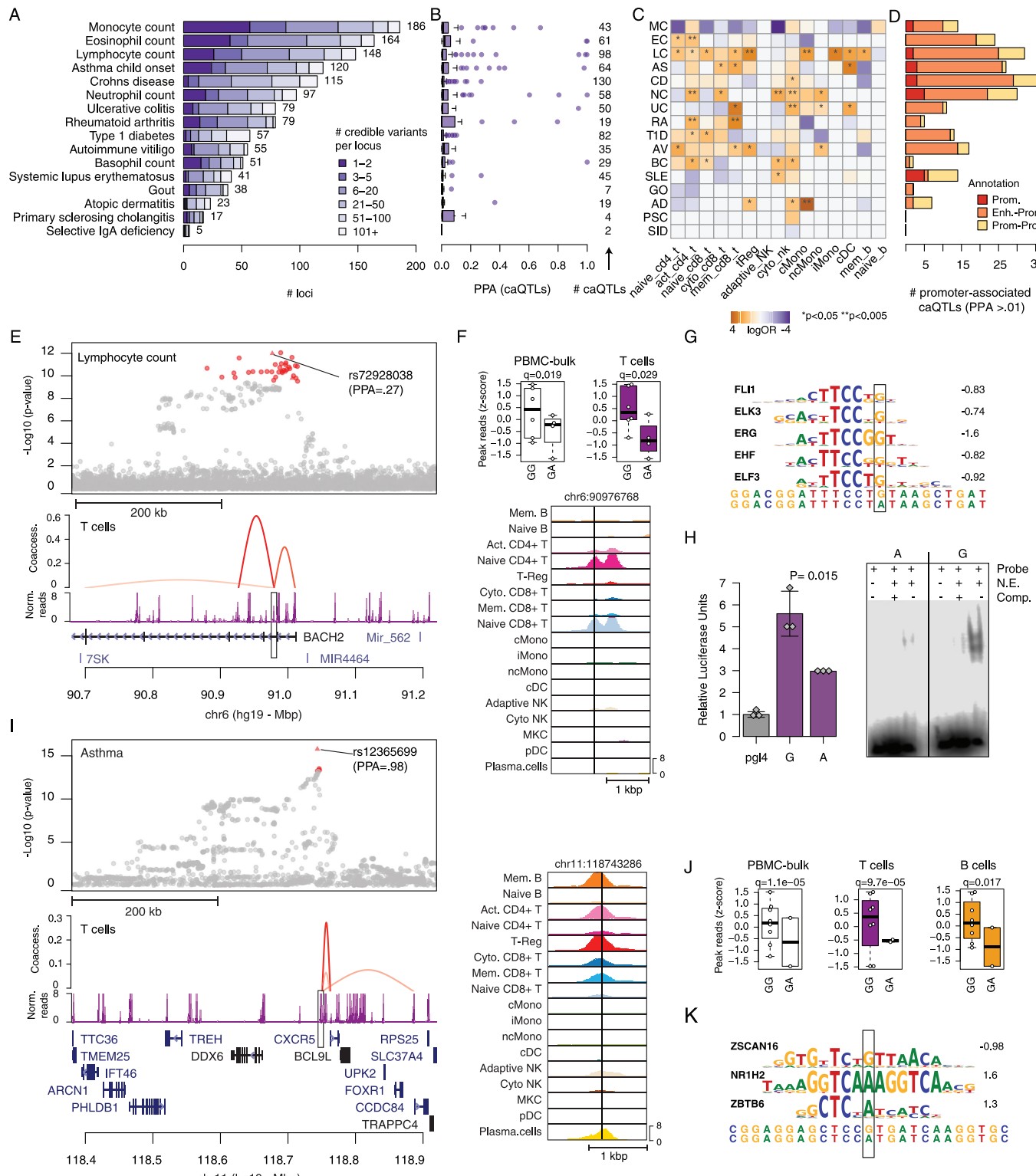

**Fig 4. Immune cell type caQTLs at fine-mapped complex immune trait loci.** A) Fine-mapping of causal variants at loci associated with 16 complex immune traits and diseases. Bar plots represent the number of credible set variants for each trait. B) Posterior probabilities of fine-mapped variants for each immune trait and disease that are caQTLs. C) Enrichment of caQTL for credible set variants for each trait. D) Breakdown of caQTLs for fine-mapped variants that were linked to promoter sites. E) Regional plot of the locus on chr6 near *BACH2*. (top) Lymphocyte count association statistics with credible set variants highlighted in red, (bottom) chromatin signal in T cells and links between the site harboring rs72928038 and co-accessible gene promoters. F) Cell type-specific caQTL effects of

rs72928038. (top) Chromatin signal by rs72928038 genotype in "bulk" PBMCs and T cells, (bottom) genome browser of chromatin signal in each cell sub-type. G) Predicted TF motifs at rs72928038, where the variant base is highlighted. H) Validation of allelic activity for rs72928038. (left) Luciferase gene reporter of sequence surrounding the G and A allele in Jurkat T cells, where the G allele had significantly higher reporter activity. (right) Electrophoretic mobility shift assay of oligonucleotides containing the G and A allele in Jurkat T cells, where the G allele had protein binding. I) Regional plot of the chr11 locus near *CXCR5*. (top) Asthma association statistics with the two credible set variants highlighted in red, (bottom) chromatin signal in T cells and links between the site harboring rs12365699 and co-accessible gene promoters. J) Cell type-specific caQTL effects of rs12365699. (top) Chromatin signal grouped by rs12365699 genotype in "bulk" PBMCs, T and B cells, (bottom) genome browser of chromatin signal in each cell sub-type. K) Predicted TF motifs at rs12365699 where the variant base is highlighted.

also a T cell-specific eQTL for both *IL6ST* and *ANKRD55* [32]. At the 2q36.3 locus associated with Ulcerative colitis, rs1811711 was fine mapped with 94% probability and was a caQTL in T cells and monocytes (S10 Table and S14B–S14E Fig) and was reported as eQTL for the nearby gene *CCL20* in whole blood.

In another example, at the 6q15 locus associated with multiple traits including lymphocyte counts (LC), type 1 diabetes (T1D) and autoimmune vitiligo (AV), rs72928038 (PPA: LC = .27, T1D = .07, AV = .01) was a caQTL in T cells ($\pi = 0.30$, q = 0.03) where the reference allele G had increased accessibility (Fig 4E and 4F). The site harboring rs72928038 was specific to naïve CD4+ T and naïve CD8+ T cells and was co-accessible with multiple gene promoters including *BACH2* (Fig 4E and S10 Table). The G allele was also predicted to have allele-specific binding to ETS family motifs, which were broadly enriched among T cell caQTLs (Fig 4G). We validated the allelic effects of this variant on regulatory activity using reporter assays in Jurkat T cells. There were significant effects on enhancer activity in luciferase gene reporter assays where the G allele had increased activity (two-sided t-test, P = .015), and allele-specific transcription factor binding to the G allele in electrophoretic mobility shift assays (Fig 4H). Previous studies have shown that this variant is a QTL in CD4+ T cells [32,33], and this site was linked to the *BACH2* promoter in promoter-capture Hi-C data in naïve CD4+ T cells [29]. Our findings provide further insight into the specificity of *cis*-regulatory effects at the *BACH2* locus across T cell sub-types and other immune cell types.

We next identified caQTLs for high-probability fine-mapped variants at loci without established molecular mechanisms. At the 11q23 locus associated with child-onset asthma, we fine-mapped a single variant rs12365699 to near-causality (PPA = .98) (Fig 4I and S10 Table). This variant was a caQTL in T cells (activated CD4+ and cytotoxic CD8+), B cells (memory B) and monocytes, where the reference and risk-increasing allele G had higher accessibility (T $\pi$ = 0.33, q = $9.7 \times 10^{-5}$, B $\pi$ = 0.32, q = 0.017) and was linked to the promoter regions of multiple genes, the closest of which was the cytokine receptor *CXCR5* (Fig 4I and 4J). The G allele of rs12365699 was also predicted to have allele-specific binding for ZSCAN16 (Fig 4K). At the 12p13.33 locus associated with lymphocyte count, we also fine mapped a likely causal variant rs34038797 (PPA = .94), which had the same strong effect in all cell type and sub-types (strongest association in classical monocytes $\pi$ = 0.20, q = $3.6 \times 10^{-5}$) and was co-accessible with multiple genes including *NINJ2* and *WNK1* (S14F–S14L Fig). The C allele had higher accessibility and higher predicted affinity with ETS transcription factors, which were ubiquitously enriched in immune cell caQTLs (S14J Fig).

## Discussion

In this study we demonstrated that profiles derived from single nucleus ATAC-seq assays of a heterogeneous tissue can be used to map chromatin accessibility QTLs in individual cell types and sub-types. While only a small number of samples were profiled in our study, we identified thousands of immune cell type and sub-type caQTLs. One reason for the larger number of caQTLs identified is that we performed QTL mapping at the level of each cell type and sub-

type, which revealed more caQTLs compared to treating each sample as a 'bulk' experiment. Another reason for the larger number of caQTLs identified was the high depth of sequencing per sample, which provided greater power particularly for allelic imbalance mapping. Supporting this, we identified substantially fewer caQTLs when down-sampling our sequence data as well as when performing population-based QTL mapping without the allelic imbalance component. As the number of unique reads covering a variant can in theory be much higher for snATAC-seq compared to bulk ATAC-seq due to having thousands of libraries per assay, the value of snATAC-seq in mapping allelic imbalance is even more pronounced.

Mapping caQTLs at cell type resolution enabled insights into cell type-specific regulation that are obscured from assays of bulk tissue chromatin. For example, we identified variants mapping in sites active in all cell types but with allelic effects on only a few cell types. We also identified variants with opposite effects on different cell types resulting in no net effect in bulk. In both scenarios, simply annotating bulk caQTLs using reference maps of cell type-specific chromatin sites would not be sufficient to uncover these effects, and therefore requires mapping accessible chromatin profiles in each cell type directly. Single cell data also enabled additional cell type-specific analyses such as linking distal sites to putative target gene promoters using co-accessibility [28]. While high-resolution maps of distal 3D interactions exist for many immune cell types in promoter-capture Hi-C [29], most other tissues do not currently have cell type-resolved interaction maps and therefore cell type co-accessibility data will be particularly valuable in annotating distal caQTLs in these tissues. Furthermore, we identified candidate transcriptional regulators through which genetic effects on chromatin accessibility in each cell type are mediated, although as TFs in the same structural sub-family often share similar motifs the specific TFs can be difficult to determine from accessible chromatin alone.

The design of our study has several notable caveats. First, there was a large difference in the number of caQTLs per cell type or sub-type. For example, we identified few caQTLs for less common sub-types such as adaptive NK cells and memory B cells, and there are even further sub-types that we did not identify. As profiling rarer cell types and sub-types using single cell assays will require a much larger number of cells per assay, cell sorting may be more efficient at present for QTL mapping in these cell types. An additional caveat was that we only tested variants in peaks for association, as opposed to all variants in a window around each site, and were unable to formally compare caQTL and disease signals. Therefore, it is possible that a subset of complex trait signals overlapping with immune cell type caQTLs have a different causal variant and may not actually affect chromatin accessibility. Colocalization analyses will be needed to establish which signals have shared causal variants with caQTLs [34]. The small sample size also constrained the classes of variants we could effectively assay to those common in European populations, and thus we had limited ability to annotate less common variants. Studies profiling larger sample sizes will circumvent this limitation, where pooling samples and deconvoluting cells into sample-of-origin [20] can help reduce per-sample cost. A further limitation is that, due to the sparsity of single cell profiles and current methods for data processing and clustering, cell type clusters will often contain misclassified and doublet cells, although this appears to have limited impact on caQTL mapping in our study. Methods such as automated labeling or single cell multimodal assays will help more accurately resolve cells [35]. Data from multimodal assays will also be valuable in defining correlative relationships between accessible chromatin and gene expression.

In summary, we identified thousands of caQTLs in immune cell type and sub-types from peripheral blood using single cell chromatin accessibility assays. Immune cell caQTLs mapped to hundreds of loci associated with complex immune traits and disease and represent a resource for interpreting the molecular mechanisms of these loci. Moving forward, mapping immune cell type-specific chromatin exposed to disease-relevant conditions and stimuli will

help further uncover a breadth of genetic effects [23,36]. These efforts will help enable more comprehensive annotation of variant function in human cell types and their contribution to complex traits and disease.

## Methods

### Ethics statement

The studies were approved by the Institutional Review Board (IRB) of the University of California San Diego. The human donors used were anonymous to the authors of this study.

### Single nuclei ATAC-seq

Peripheral blood mononuclear cells (PBMCs) from 13 individuals (7 females and 6 males) were purchased from HemaCare (Northridge, CA) and profiled for snATAC-seq using 10x Genomics Chromium Single Cell ATAC Solution, following manufacturer's instructions (Chromium SingleCell ATAC ReagentKits UserGuide CG000209, Rev A) as described previously [17]. Briefly, cryopreserved PBMC samples were thawed, resuspended in 1 mL PBS (with 0.04% FBS) and filtered with 50 μm CellTrics. Cells were centrifuged and permeabilized with 100 μl of chilled lysis buffer (10 mM Tris-HCl pH 7.4, 10 mM NaCl, 3 mM MgCl2, 0.1% Tween-20, 0.1% IGEPAL-CA630, 0.01% digitonin and 1% BSA) for 3 min on ice and then washed with 1mL chilled wash buffer (10 mM Tris-HCl pH 7.4, 10 mM NaCl, 3 mM MgCl2, 0.1% Tween- 20 and 1% BSA). After centrifugation, pellets were resuspended in 100 μL of chilled Nuclei buffer (2000153, 10x Genomics) in a final concentration of 3,000 to 7,000 of nuclei per μl. 15,300 nuclei (targeting 10,000) were used for each sample. Tagmentation was performed using nuclei diluted to 5 μl with 1X Nuclei buffer, 10x ATAC buffer and ATAC enzyme from 10x Genomics, for 60 min at 37˚C. Single cell ATAC-seq libraries were generated using the Chromium Chip E Single Cell ATAC kit (10x Genomics, 1000086, PBMC1 to 12) or Chromium Next GEM Chip H Single Cell Kit (10x Genomics, 1000161, PBMC13 to 15) and indexes (Chromium i7 Multiplex Kit N, Set A, 10x Genomics, 1000084) following manufacturer instructions. Samples were sequenced to an average depth of 212 million 50-nt read pairs each, using Illumina HiSeq4000 at the UCSD Institute for Genomic Medicine. Alignment to the hg19 genome and initial processing were performed using the 10x Genomics Cell Ranger ATAC v2.0 pipeline. Sample information and a summary of the Cell Ranger ATAC-seq quality metrics are provided in S1 Table. We also used publicly available snATAC-seq datasets in PBMCs provided by 10x Genomics (S2 Table) to help further improve clustering and cell type/ sub-type identification, which were processed using the same pipeline.

### Quality control, clustering, and cell type assignment

To generate a matrix of reads per nucleus in each sample, the BAM files were first filtered for reads with MAPQ<30, secondary or unmapped reads, and duplicate reads using samtools [37]. BAM files were then converted into tagAlign (bed) files, where reads were shifted by +4 bp (positive strand) or by -5bp (negative strand), extended to 200 bp, and centered. We then generated a bed file with the coordinates of hg19 split into 5 kb windows, filtered for windows overlapping blacklisted regions from ENCODE (version 2), and intersected it with the tagAlign file of snATAC-seq reads. This intersection file was then used to create a sparse $m$ x $n$ matrix containing read depth for $m$ nuclei and $n$ 5kb-windows using the R package matrix.

For each sample, we retained nuclei with a minimum read count of 4,000. After this initial pre-filtering step there were on average 9.1k nuclei per sample (S2 Table). Individual sample matrices were then concatenated using python AnnData package (117,824 total nuclei). Using

scanpy [38] (v.1.5), we extracted highly variable windows using mean read depth and normalized dispersion (min_mean = 0.01, min_disp = 0.25). After normalization to uniform read depth and log-transformation, we regressed out the log-transformed read depth within highly variable windows for each cell. The "pca" function in scanpy was then used to perform principal component analysis. The top 50 principal components were extracted and corrected to remove batch effects using harmony [39] version 0.1.0 (arguments "theta = c(2,2) and "do_pca = FALSE"), using sample donor ID as a covariate. The harmony-corrected components were then used to calculate the nearest 30 neighbors using the cosine metric. This cosine metric was then used to perform UMAP dimensionality reduction clustering with the parameters 'min_dist = 0.3', and Leiden clustering [40] (resolution = 1.5). To filter nuclei with low quality, entire clusters with low fraction of reads in promoters, low log usable read count, low fraction of reads in peaks, and/or clusters that did not have uniform representation across samples were iteratively removed across multiple rounds of clustering. A total of 15,958 nuclei were removed by this approach. We then performed sub-clustering of each cluster using the Louvain algorithm (resolution = 2) and removed sub-clusters within each cluster with higher read counts or accessibility for multiple cell type marker genes (n = 5,864 nuclei removed). At the final round of clustering, we removed the nuclei from 10x public datasets at just retained the nuclei in the 13 PBMC samples in our study. After all filtering steps and final clustering with 13 PBMC samples, the final map consisted of 96,002 nuclei mapping to 17 clusters. There were on average 7.4k nuclei per sample in the final map, a 20% reduction from the starting number of pre-filtered nuclei (S1 Table). To then assign cell type and sub-type identity to each cluster, we used chromatin accessibility at 5kb genomic windows overlapping promoter regions of known marker genes (S3 Table).

## Comparison with scRNA-seq data

To verify the cell type and sub-type labels of clusters assigned using snATAC-seq, we compared genes with accessible chromatin profiles specific to each cluster to genes with cluster-specific expression from single cell RNA-seq data in PBMCs [21]. For each of the 17 snATAC-seq clusters, we used the 'rank_gene_groups' function in scanpy (with the arguments 'method = wilcoxon' and 'corr_method = benjamini-hochberg') to identify cluster-specific windows of accessibility in each cluster with respect to all other clusters. For each cluster we then extracted a list of the top 100 z-score ranked promoter-annotated windows (aka genes) that had a minimum of 1.3 log fold change with respect to the other clusters. The same analysis was performed on a published scRNA dataset [21] to identify the top 100 specific genes in the corresponding clusters. The intermediate monocyte cluster was excluded because it was not annotated in the scRNA dataset. Using a merged list of these cluster-specific genes (1,792 in total), we then performed Pearson correlation between gene expression and promoter accessibility z-scores in each pair of clusters.

## Peak calling

For each cluster, mapped reads were extracted from all cells within the cluster. Reads aligning to the positive strand were shifted by +4 bp and reads aligned to the negative strand were shifted by -5 bp. Reads were extended to 200 bp and then centered, and bed files were created from the resulting read coordinates. We then called peaks with MACS2 [22] from the bed files using the parameters '-q 0.05', '-nomodel', '-keep-dup all', '-g hs', and '-B'. The read count pileup bedgraph was sorted and normalized to counts per million (CPM), converted to bigwig and visualized using the UCSC Genome Browser. We created a merged peak set by combining narrow peak files across all clusters into a single bed file using bedtools merge.

## Comparison with bulk immune cell ATAC-seq data

We obtained published data of FACS-sorted immune cell types (GSE118189) [23], mapped reads to hg19 using bwa mem [41] and removed duplicate reads. We merged replicate samples and performed peak calling for each cell type as described above. Mapped reads from immune cell types and sub-types derived from snATAC-seq in this study and from the bulk immune cell ATAC-seq profiles were used to generate bedgraph files using bedtools [42]. Read counts were normalized to CPM and bigwig files were generated using ENCODE 'bedgraphToBig-Wig' [43]. We created a bed file of the union of peak calls from snATAC-seq and bulk ATAC-seq using bedtools. We then compared bulk ATAC-seq cell type and snATAC-seq cell type normalized read count profiles within the union peak set using deeptools 'multiBigWigSummary' [44]. A heatmap of the clusters of Spearman rank correlation coefficients indicating similarity between bigwig files was generated using the summary comparison from 'multiBigWigSummary'.

## Sample genotyping and imputation

Genomic DNA form PBMC samples was extracted using the PureLink genomic DNA kit (Invitrogen). Genotyping was performed using Infinium Omni2.5–8 v1.4 and v1.5 arrays (Illumina) at the UCSD Institute for Genomic Medicine. Genotypes were assigned with GenomeStudio (v.2.0.4) with default settings, using manifests for the human reference genome built 38, and exported using the PLINK plugin. Variants were prepared for imputation using the perl script "HRC-1000G-check-bim.pl" v4.3.0 (https://www.well.ox.ac.uk/~wrayner/tools/), with the TOPMed reference panel hg38, freeze 8 (https://bravo.sph.umich.edu/freeze8/hg38/). Variants with ambiguous alleles (G/C, or A/T) and MAF > 0.4, variants with differing alleles, variants with >0.4 allele frequency difference and variants not in reference panel were removed. The remaining variants (n = 1,325,601) were uploaded to the Michigan Imputation Server to impute genotypes using the TOPMed r2 panel, with minimac4. We then retained variants with imputation quality R2>0.3 (n = 9,087,555). After imputation, the genomic coordinates of the variants were lifted-over from hg38 to hg19 using picard LiftoverVcf (http://broadinstitute.github.io/picard/) and variants that could not be mapped or that mapped to different chromosomes after lift-over were removed.

To confirm the ancestry of PBMC donors, we compared genotypes to 2,504 samples in the 1000 Genomes Project Phase (1KGP) 3 (https://www.internationalgenome.org/). Genotyped variants (before imputation) were lifted-over to hg19 and merged with the 1KGP genotypes using PLINK [45], retaining only variants in common with the two groups. Duplicated variants, variants with MAF <0.01 and non-independent variants (—indep 50 5 2) were filtered using PLINK. Principal component analysis (—pca) was performed using PLINK on the remaining variants (n = 316,506).

## Removing mapping bias

We used WASP [24] to remove mapping bias to the reference genome at polymorphic sites. The recommended WASP pipeline (https://github.com/bmvdgeijn/WASP) was modified to perform each step on a per barcode basis. For each sample, we split the Cell Ranger ATAC alignments (possorted BAM) into individual alignments for each cell barcode (only using barcodes that remained after QC filtering). Then, for each alignment, the WASP script "find_intersecting_snps.py" was applied to extract reads mapping to a list of SNPs and INDELS from the imputed genotype that intersected a snATAC-seq consensus peak. Reads overlapping to SNPs were then re-mapped using BWA MEM and processed with samtools fixmate and samtools sort. Reads overlapping to INDELS were removed from the alignments by the program.

Subsequently, the WASP command "filter_remapped_reads.py" was used to create a list of reads that remapped correctly to the same genomic position, which were then merged with the ones that did not need to be re-mapped, and then sorted. Duplicate reads were then removed using the WASP command "rmdup_pe.py", followed by a final merging step combining all filtered individual barcode-alignments for each sample. For the "bulk" assay, the same WASP pipeline was used on the original 10x 'possorted' BAM files without splitting into different barcodes.

## Identification of chromatin accessibility QTLs

For each sample of European genetic ancestry (10 samples total, excluding samples PBMC1, PBMC6 and PBMC15, S1 Fig), we split reads in the WASP-filtered snATAC-seq.bam files according to cluster label. For each cell type and sub-type cluster, we generated peak count matrices (peak x sample) using merged peak site coordinates and the split.bam files using featureCounts [46]. We then obtained VCF files of SNPs located within peaks and annotated allelic read counts using RASQUAL tools [25]. We filtered for variants heterozygous in at least 2 samples. For the "bulk" experiment we ignored cell type labels and barcodes and used all reads.

For each cell type and sub-type, we retained only accessible sites with at least 5 reads on average across samples. To perform caQTL analysis we used RASQUAL and tested for association between each peak and variants contained in the peak itself or in other peaks within a +/-10Kb window. We included the library size of each sample calculated using the rasqualCalculateSampleOffsets() function and read count covariates using make_covariates() function in each model. The number of ATAC-seq read count covariates were dynamically calculated for each cell type and sub-type and therefore different cell types/sub-types had different numbers of covariates. We also included the first four PCs derived from genotype data together with major 1KG populations as covariates in each model.

For each peak, RASQUAL calculated adjusted p-values accounting for the number of variants tested per peak, and the variant with the minimum adjusted p-value was marked as the lead variant. To correct for multiple testing genome-wide, we performed 2 additional association tests with permuted genotypes (-r option in RASQUAL) and calculated an empirical FDR (10%) by comparing the q-values of the real and permuted association results. To control for potential genotyping error, we identified and excluded caQTLs with reference allele mapping bias ($\psi$) $<0.2$ or $>0.8$ and those with sequencing mapping error rate ($\partial$) $>0.1$.

To assess the effect of misclassified or residual doublet nuclei in each cluster on caQTL mapping, we first determined the proportion of nuclei with at least one cell with promoter accessibility for several highly cell type-specific genes in each of the major cell types (B-cells, T-cells, NK-cells, Monocytes). For each cell type, we removed nuclei with accessibility at the promoter of genes specific to different cell types (B cells n = 150, T cells n = 1,213, NK cells = 299, Monocytes n = 2,007), and repeated caQTL mapping using the remaining nuclei. We compared results by determining the correlation in effect sizes and concordance in direction of effect for caQTLs.

To assess the impact of sequencing depth on caQTL identification, the WASP-filtered 'bulk' alignments were subsampled to 25% of depth using 'samtools view -s 0.25' and caQTL analysis was repeated.

To confirm that caQTLs were not biased for genotypes with only two observed classes (homozygous major and heterozygous) over genotypes with three observed classes (homozygous major, heterozygous, homozygous minor), for each cell type we stratified the lead variants from summary statistics based on the two genotypes categories and i) visualized p-value distributions with Q-Q plots and ii) calculated enrichment for caQTLs in 3-genotype vs 2-genotype

classes using a Fisher's exact test. We also calculated the enrichment of caQTLs for each genotype combination in the 10 individuals (S8 Fig).

### caQTL detection sensitivity

To estimate the sensitivity of caQTL detection across different cell numbers, we sampled 10, 25, 50, 100, 250, 500, 1k and 2k cells per individual from the cell population with highest cell count, classical monocytes (1,750.5 on average per sample, 10 samples). Barcodes were randomly sampled (with replacement in the case of 2k cells per individual) and used to subset classical monocyte BAM files for each sample. We then performed caQTL mapping using the exact same method as for regular cell sub-types. We determined sharing in each sub-sample with the original caQTLs by calculating the π1 statistic using the *qvalue* package in R [47] as described by Chen et al [15] with method = "bootstrap". We also calculated sharing between cell type and sub-type caQTLs and 'bulk' caQTLs using the same π1 statistic. We note that the assumption of the method that p-values are uniformly distributed is not met by the RASQUAL results.

### Compare caQTLs between cell types and with external datasets

To estimate the correlation of effect sizes of caQTLs across cell types and "bulk" data we calculated the spearman correlation coefficient of effect sizes (π) in each pair of cell types and "bulk". For each comparison we selected SNP-peak pairs that were significant caQTLs in at least one of the two cell types. Correlation coefficients were tabulated in a matrix and hierarchically clustered using 'pheatmap'. The "bulk" and cell type caQTLs were compared with caQTLs from 24 LCLs, also calculated using RASQUAL [25]. For each dataset we intersected the peaks tested in PBMCs with those tested in LCLs and estimated enrichment for being caQTLs in both datasets using Fisher's exact test. To estimate coordination of caQTLs effects we selected PBMC peaks with variants that were tested in the LCL study (only lead variants were available) and were caQTLs in either PBMCs or LCLs and calculated the fraction of concordant effects. Monocyte and T-cells single-cell caQTLs were compared with H3K27ac QTLs and eQTLs from FACS sorted Monocytes and T-cells from the BLUEPRINT project, calculated using WASP and the Combined Haplotype Test at FDR 10%, which similarly to RASQUAL considers both allelic and population effects. For each comparison we selected variants tested in both datasets and calculated enrichment for shared variant QTLs (lead variants only) using Fisher's exact test.

To identify caQTLs with evidence of cell type specific effects we performed an interaction test following a previously described method [48]. For each resolution (cell type or subtype), we created a count matrix with read counts for the consensus peaks in each sample in each cell type or subtype. For example, the count matrix for major cell types had 40 columns (10 samples x 4 cell types). We then performed variance-stabilized normalization using the vst() function from DESeq2 to obtain normalized read counts. We extracted the four principal components from the normalized read count matrix to use as covariates, together with age, sex and four principal components from genotyping data. We then selected all unique peak-lead variant pairs caQTLs in any cell type at FDR 10% and tested if they were significantly different between cell types. For each peak-lead variant, we created linear models of expression level as a function of genotype and cell type with or without an interaction term between cell type and genotype as well as age, sex, three PCs derived from genotypes, and four PCs derived from accessible chromatin profiles. We then compared models with and without an interaction term using ANOVA and obtained the p-value and used Benjamini-Hochberg correction of the p-values to obtain a list of peak-lead variants with evidence of heterogeneity at FDR<10%.

## Transcription factor motif analysis

To identify enriched motifs that were altered by caQTLs we used the package MotifBreakR [26]. First, we selected 152,722 SNPs that were tested in any of the cell types for caQTLs and imported them using the function snps.from.file(), using hg19 as reference genome. Then we determined if they disrupted TF motifs from the HOCOMOCO v10 human database [27], comprising 640 motifs corresponding to 595 unique TFs, and accessed via MotifDb. The following motifbreakR() function parameters were used: filterp = TRUE, method = "ic", = 5e-4, BPPARAM = BiocParallel::bpparam("SerialParam"). SNPs that resulted in disruption of any TF motif with a *strong* effect (defined by motifbreakR) were considered as motif altering (n = 145,340). To calculate enrichment for alteration of specific TF in caQTLs of major cell types (B-cell, T-cell, NK_cell and Monocyte), we performed a one-tailed exact binomial test (binom.test(alternative = "greater")) comparing the frequency of alteration of a motif by caQTLs to the total frequency of motif alteration in the tested SNPs for each cell type. Significant enrichment was considered at a Benjamini & Hochberg corrected P-value<0.05. To display enriched motifs, we used the packages MotIV and motifPiles, selecting the top motifs in each cell type ranked by p-value.

To determine which TF genes may underlie sequence motifs enriched for caQTLs in each cell type, we used a published scRNA dataset in PBMCs [21]. For each of the 4 major cell types (monocyte, T cell, B cell, NK cell) in scRNA-seq data, we applied the 'rank_gene_groups' function in scanpy with the arguments 'method = wilcoxon' and 'corr_method = benjamini-hochberg'. We identified TF genes preferentially expressed in each cell type cluster with respect other clusters with FDR<0.05 and log fold change >1, and matched TFs genes to corresponding TF sequence motifs.

## Single cell co-accessibility

Peak-to-peak co-accessibility was calculated using Cicero (version 1.1.5) [28] for B cells, T cells, NK cells, and Monocytes. We created a long format matrix encoding the snATAC-seq barcodes for each cell in each cell type and the superset of ATAC-seq peaks for all cell types, indicating which cells were accessible in which peaks. For each cell type, the cicero function 'make_cicero_cds' was used to aggregate cells into bins of 30 nearest neighbors (parameter k = 30) from the UMAP reduced dimensions obtained from clustering. We then calculated co-accessibility scores using a window size of 1 Mb. Once co-accessibility scores were calculated, a threshold of 0.05 and a minimum distance of 10 kb were used to define pairs co-accessible for a given cell type. We used a 0.05 threshold as this has been previously shown in other snATAC-seq studies to provide strongest enrichment for 3D chromatin interactions [17,28,49], and also showed strongest enrichment for immune cell 3D interactions in our study (S13A Fig). We further generated cell type co-accessibility for each sample individually and filtered sites co-accessible at .05 in less than four individual samples. A peak was categorized as 'promoter' if it fell within a 2 kb window of a transcription start site based on GENCODE (version 19) promoter annotations [50], and otherwise was categorized as 'distal'.

To validate the cell-type specificity of promoter-distal links calculated using co-accessibility, we compared them to chromatin interactions from promoter capture Hi-C (pCHi-C) data previously generated in 16 immune cell types and sub-types [29]. We obtained the list of promoter baits and the matrix containing CHiCAGO scores for all interactions in all immune cell type. First, for each pair of peaks that we analyzed in each cell type, we filter those where at least one peak intersected (+/- 1kb) a pCHi-C bait using pgltools [51]. Then, we identified overlapping connection between the filtered pairs of sites in each of our 4 cell type (B-cells, T-cells, NK cells, and Monocytes) and the pCHi-C connection (CHiCAGO score > = 5) from each of the

16 blueprint adult cell types (Mon, Mac0, Mac1, Mac2, Neu, MK, EP, Ery, nCD4, tCD4, aCD4, naCD4, nCD8, tCD8, nB, tB) using the function compare_connections() from the Cicero package, and using a maximum gap of 1,000 bp allowed. For each celltype-celltype comparison we then estimated the enrichment for the co-accessible sites in pCHi-C connection using Fisher's exact test. Odds ratios for each comparison were tabulated and displayed using heatmap. For each of the best-matching cell types (B-cells-tB, T-cells-tCD8, and Monocytes-Mon, NK-cells-tCD8) we also calculated enrichment at different peak distances (10–50, 50–100, 100–200, 200–350, 350–1000 kb).

## Distal effect of caQTL variants on co-accessible promoters

To examine the effect of caQTLs on co-accessible sites, for each of the 5 major cell types we took the lead caQTL variant and tested for association with accessibility level of the co-accessible site using RASQUAL [25]. We used the same method as above for caQTLs except for adopting a more relaxed FDR threshold of 20% instead of 10%. caQTL-coaccessible peaks were then filtered to retain only enhancer-promoters and promoter-promoter co-accessible peaks (B-cells n = 1,639, T-cells n = 6,080, NK cells n = 2,146, Monocytes n = 4,494, with an average number of 3.45 co-accessible promoters for each caQTL). The pearson correlation of effect sizes was calculated between variant effect on the original caQTL peak and on one of the co-accessible promoters (with lowest RASQUAL p-value of association), and only considering co-accessible peaks at >10kb of distance.

## Genetic fine mapping analysis

We obtained genome-wide summary statistics for immune-related phenotypes including blood cell type counts [52], autoimmune diseases [53–57], and inflammatory diseases [58–60]. For each study, we obtained lists of index variants for each independent signal from the supplement. We used PLINK [45] to estimate linkage disequilibrium (LD) between these index variants and all variants within ±2.5 Mb using samples of European ancestry from the 1000 Genomes Project [61]. For each signal, we first pre-filtered variants in at least low LD ($r^2>0.1$) with the index variants. We calculated approximate Bayes factors [62] (aBF) for each variant using the effect estimates (β) and standard errors (SE), assuming prior variance w = 0.04. We calculated the posterior probability of association (PPA) by dividing the aBF for each variant by the sum of aBFs for all variants included in the signal. We then defined the 99% credible set as the smallest set of variants that added up to 99% PPA. Fine-mapped variants were annotated using cell type and sub-type caQTLs, considering each lead variant as well as variants with the same q-value of the lead variant for each caQTL. Fine-mapped caQTL variants with PPA>1% were then further annotated with co-accessible promoters (S10 Table).

To test for enrichment of caQTLs for complex immune traits we calculated the cumulative PPA of variants overlapping immune cell sub-type caQTL peaks across all credible sets for each trait. For each cell sub-type, we defined a background set of peaks tested for association but did not have significant caQTLs. We estimated an empirical distribution for the total PPA using 1,000 random draws of peaks from the background equal in number to the caQTL sites. For each test (trait vs cell sub-type) a p-value was calculated by comparing the total PPA within caQTL peaks to the empirical distribution.

## Luciferase gene reporter assays

Human DNA sequences (Coriell) with reference allele for rs72928038 (*BACH2* intron) were cloned in forward orientation in the luciferase reporter vector pGL4.23 (Promega) using the primers: forward, AGCTAGGTACCACACTCAGTGGTTGGGGTTT, and reverse,

TACCAGAGCTCCTGGATAGAGGTCCCAGTCG and the enzymes SacI and KpnI. Alternate allele plasmids were generated via site directed mutagenesis (Q5 SDM kit, New England Biolabs) using the following primers: forward, CGGATTTCCTaTAAGCTGATC, reverse, TCCCTATTTGTGTGTAATG.

Jurkat cells were maintained in culture at a concentration of $1\times10^{05}$/mL-$1\times10^{06}$/mL. Approximately $0.5\times10^{06}$ cells per replicate (3 replicates) were co-transfected with 500 ng of firefly luciferase vector containing either the reference or alternate allele or an empty pGL4.23 vector as a control, and 50 ng pRL-SV40 Renilla luciferase vector (Promega), using the Lipofectamine LTX reagent. Cells were collected 48 hours post transfection and assayed using the Dual-Luciferase Reporter system (Promega). Firefly activity was normalized to the Renilla activity and expressed as fold change compared to the luciferase activity of the empty vector (RLU). A two-sided t-test was used to compare the luciferase activity between the two alleles.

## Electrophoretic mobility shift assays

EMSAs were performed according to manufacturer's instruction, with changes indicated below, using the LightShift Chemiluminescent EMSA Kit (Thermo Scientific, 20148). Biotinylated and non-biotinylated single-stranded oligonucleotides harboring the rs72928038 variant (5'-TAGGGACGGATTTCCTGTAAGCTGATCTTGAAG-3', 5'-TAGGGACGGATTTCC-TATAAGCTGATCTTGAAG-3') were purchased from Integrated DNA Technologies. Nuclear extract from E6-1 Jurkat T cells (ATCC TIB-152), cultured as described above, was obtained using the NE-PER Nuclear and Cytoplasmic Extraction Reagents (Thermo Scientific, 78833). Binding reactions were carried in a total volume of 20 μl, with 10X Binding Buffer (100 mM Tris pH 7.5, 500 mM KCl and 10 mM DTT), 2.5% glycerol, 5 mM MgCl2, 0.05% NP40, 50 ng Poly(dI:dC), 100 fmole of biotin-labeled probe, and 5.1 μg nuclear extract. For competition experiments, 20 pmol of unlabeled probe was added. Competition reactions were incubated at room temperature for 10 mins before the addition of the biotin-labeled probe. At the addition of the biotin-labeled probe, all reactions were incubated at room temperature for 20 min. Reactions were loaded onto a 6% polyacrylamide 0.5X TBE Gel (Invitrogen, EC62655BOX) for electrophoresis and transferred for 45 mins to a Biodyne B Pre-Cut Modified Nylon Membrane, 0.45μm (Thermo Scientific, 77016). Transferred DNA was UV-cross-linked for 15 mins, and the biotinylated probes were detected using Chemiluminescent Nucleic Acid Detection Module (Thermo Scientific, 89880) following the manufacturers instruction, with initial blocking increased to 60 mins. The image was acquired using C-DiGit Blot scanner (LI-COR Biosciences, Model 3600).

## Supporting information

**S1 Fig. Population structure of PBMC samples.** The first six principal components derived from joint analysis of genotype data from the 1000 Genomes Project and PBMC samples. Samples in 1000 Genomes are colored by major population group, and the PBMC samples are colored in pink, and further shown in separate plots on the right.
(TIF)

**S2 Fig. Defining immune cell types and sub-types from snATAC-seq.** A) UMAP plots showing promoter accessibility in a 1 kb window around the TSS for selected cell type marker genes (see S3 Table). B) Proportion of nuclei in each cell type with at least one read mapping to a promoter peak for cell-type specific genes. The percentage of nuclei with accessible marker promoter with respect to the correct cell type is shown on top of each bar. C) Heatmap

showing Pearson correlation between the top cluster-specific genes in scRNA-seq and snA-TAC-seq clusters using a Wilcoxon signed-rank test.
(TIF)

**S3 Fig. snATAC-seq profiles at marker genes.** Genome browser plots showing aggregate read density (scaled to $1\times10^5$ read depth, y-axis: 0–8 normalized read depth) for cells within each cell type for selected cell type marker genes.
(TIF)

**S4 Fig. Immune cell type snATAC-seq profiles in individual PBMC samples.** A) UMAP plot showing cells in each of the 13 PBMC samples assayed in this study. B) Scatter plot comparing cell type proportions obtained from cluster analysis versus those obtained from flow cytometry, excluding leukocytes. Proportions represent the fraction of all cells in each sample (see S4 Table for individual sample proportions). Each dot represents an individual sample. C) Barplot showing the number of cells assigned to 17 distinct immune cell types and sub-types in each sample. D) Barplot showing the relative proportion of cells from each sample in each immune cell type and sub-type.
(TIF)

**S5 Fig. Comparison of ATAC-seq peaks from PBMC snATAC-seq and FACS sorted PBMCs.** Heatmaps and hierarchical clustering of Spearman correlation coefficients for pair-wise comparisons of genome-wide ATAC-seq profiles across cell sub-types from PBMC snATAC-seq from this study (columns) and from a published bulk ATAC-seq study of FACS sorted immune cells (rows). An asterisk denotes the best matching (highest correlation) cell type from snATAC-seq to FACS-sorted immune cells.
(TIF)

**S6 Fig. Summary of caQTLs in immune cell types and subtypes.** A) Number of caQTLs identified in each cell type, sub-type, or "bulk" resolution, at different FDR thresholds. B) Q-Q plots of nominal p-values (one top variant per peak) for each caQTL dataset. C) Scatterplot showing correlation between number of cells per cell type and number of caQTLs identified. D) Comparison of effect sizes (converted to effect allele direction) at shared sites between "bulk" (black) and major cell types (color) caQTLs. P-value of two-sided paired Wilcoxon test is shown.
(TIF)

**S7 Fig. Effect of coverage on caQTL identification.** Boxplots for cell types, sub-types, and "bulk" show the distribution of mean read depth for all peaks, tested peaks, and significant caQTL peaks in either all samples or only heterozygous samples. The median of the distribution is indicated in green. The caQTLs (last 2 boxplots) have higher median coverage compared to tested peaks. The histograms on the bottom show the average read depth for caQTLs, with the number of peaks with lower coverage (mean <10 reads) shown in red.
(TIF)

**S8 Fig. Effect of genotype classes on caQTL identification.** A) Distribution of genotype classes across all tested variants indicated as number of individuals with homozygous reference, heterozygous and homozygous alternate alleles. Two- and three-genotype classes are color-coded by red and black, respectively. B) Fisher's exact test results showing enrichment for three-genotype classes in caQTLs in each cell type, sub-type, or "bulk". C) Fisher's exact test enrichment results for individual genotype classes in B cells, T cells, Monocytes and NK cells. D) Q-Q plots of lead SNPs stratified between two- (red) and three- (black) genotype classes.
(TIF)

**S9 Fig. Comparison of caQTL effects with and without the allelic imbalance component.** For each cell type (first 4 plots) and sub-type (remaining 15 plots), a scatter plot shows the consistency between caQTL effect ($\pi$) in the standard QTL analysis including allelic imbalance (x-axis,) and the effect for the same variant-peak pair excluding the allelic imbalance component (y-axis). The latter was obtained by running RASQUAL using the—population-only option. Black dots indicate caQTLs that were significant only in the standard analysis, and red dots indicate caQTLs that were significant in both analyses. The percentage of discordant effects are indicated.
(TIF)

**S10 Fig. Comparison of cell type-specific and shared caQTLs.** A) Properties of caQTLs shared by more than one cell type and present only in that cell type, as well as non-caQTLs (NS), for the four major cell types (B cells, T cells, Monocytes, NK cells). MAF = minor allele frequency within PBMC samples, abseffect = "absolute" (effect allele) effect size. P-values are from a two-tailed t-test. B) Number of caQTLs with cell-type specific effects based on ANOVA and classified by whether the caQTL is for a cell type-specific or cell type-shared peak.
(TIF)

**S11 Fig. Comparison of snATAC-seq caQTLs with LCL caQTLs from 24 individuals.** A) Significant caQTLs in Monocytes, T-cells, NK-cells and B-cells and their overlap with LCL caQTLs, considering only peaks tested in both datasets and where the same variant was tested. For caQTLs identified in either study (indicated in the legend), a scatter plot shows effect sizes for each study. B) P-values and odds ratio from a two-tailed Fisher's exact test for enrichment of cell type caQTLs in LCL caQTLs.
(TIF)

**S12 Fig. Examples of caQTLs specific to immune sub-types.** A) Examples of cell-type specific caQTLs due to presence of the peak in a single cell type. From left to right, rs1957554 is a caQTL for a naïve B cell-specific site, rs855166 is a caQTL for a naïve CD4+ T cell-specific site, rs2459596 is a caQTL for a classical monocyte-specific site, and rs59176853 is a caQTL for a cytotoxic NK cell-specific site. Top panels: color-coded boxplots show association in the different cell types, white boxplots show corresponding caQTL in "bulk" PBMCs. Association q-values are shown on the top and variant genomic location (hg19) is shown at the bottom. Bottom panels: genome-browser screenshot of snATAC-seq in different cell types.
(TIF)

**S13 Fig. Correlation of caQTL effects with effects on distal promoter sites.** A) Enrichment of co-accessible sites in promoter-capture Hi-C loops at different cicero score thresholds. B) Comparison of distributions for promoter-capture Hi-C loops (red) or Cicero interaction (blue) across different distance bins. C) Correlation in the effects of caQTL variants on the QTL site and co-accessible promoter sites in each cell type, grouped by distance between the QTL site and co-accessible promoter site. Pearson correlation coefficient and number of co-accessible pairs of peaks are indicated.
(TIF)

**S14 Fig. Additional examples of immune cell type caQTLs at fine-mapped complex immune trait loci with high causal probability.** A) Minor allele frequency in 1000 Genomes EUR samples of fine-mapped variants tested for caQTL association in this study. B) Regional plot of the *CCL20* locus on chr12 associated with ulcerative colitis, with credible set variants highlighted in red. Candidate causal variant rs1811711 is indicated with a triangle. C) Chromatin signal in T cells and Monocytes at the same locus and co-accessibility between the site

harboring rs1811711 and *DAW1*, *AGFG1*, *COL4A3* and *COL4A4* promoters. D) Chromatin signal grouped by rs1811711 genotype in "bulk" PBMCs, Monocytes and T cells and q-values for caQTL association. E) Zoomed-in genome browser track of the caQTL peak in each sub-type. F) Regional plot of the locus on chr12 in the *NINJ2* gene showing association with lymphocyte count, with credible set variants highlighted in red. Candidate causal variant rs34038797 is indicated with a triangle. G) Chromatin signal in T cells and Monocytes at the same locus and co-accessibility between the site harboring rs34038797 and *CCDC77*, *WNK1*, *RAD52*, *NINJ2* and *SLC6A12* promoter. H) Chromatin signal grouped by rs34038797 genotype in "bulk" PBMCs and all major cell types and q-values for caQTL association. I) Zoomed-in genome browser track of the caQTL peak in each sub-type. J) Predicted TF sequence motifs altered by rs34038797, where the variant base is highlighted.
(TIF)

**S1 Table. Summary of PBMC samples.** Donor characteristics for each PBMC sample.
(XLSX)

**S2 Table. Summary of snATAC-seq data.** Sequencing and mapping statistics of snATAC-seq assays for each PBMC sample.
(XLSX)

**S3 Table. Marker gene references.** List of marker genes used to assign clusters to PBMC cell types and sub-types and corresponding references.
(XLSX)

**S4 Table. Clustering vs. flow cytometry cell type proportions.** Comparison between cell type proportions in each sample as estimated by flow cytometry and snATAC-seq.
(XLSX)

**S5 Table. Immune cell type and sub-type accessible chromatin sites.** Merged bed file of all ATAC peaks sites called in each cell sub-type, used for all analyses.
(XLSX)

**S6 Table. Immune cell type and sub-type caQTLs.** All caQTLs significant at FDR 10% in each cell type (4 major cell types, 17 sub-types) and in"bulk". Only the lead variant for each caQTL is included. The first 25 columns are outputs from RASQUAL, and p-values and q-values were calculated from columns 11 and 10, respectively.
(XLSX)

**S7 Table. Sensitivity of caQTL detection.** Estimated sharing of caQTLs between cell-type and sub-type caQTLs and 'bulk' caQTLs, as well as between caQTLs in sub-sampled classical monocyte cells compared to caQTLs using all classical monocyte cells.
(XLSX)

**S8 Table. Transcription factor motifs enriched in immune cell type caQTLs.** List of TF motifs from the HOCOMOCO v.10 human database that were tested for enrichment in caQTLs with results of binomial test and cluster-specific TF gene expression from scRNA-seq.
(XLSX)

**S9 Table. Complex immune traits and diseases included in fine-mapping.** List of complex immune traits and corresponding GWAS studies used for fine mapping.
(XLSX)

**S10 Table. Immune cell type and sub-type QTLs at fine-mapped variants.** Credible set variants for complex immune traits and diseases that are caQTLs in one or more cell type, have

PPA >0.01, and are located either in distal enhancers co-accessible with a gene promoter or in a gene promoter directly. For each fine-mapped variant we report caQTL results and co-accessible promoters in each cell type with significant caQTLs.
(XLSX)

## Author Contributions

**Conceptualization:** Kyle J. Gaulton.

**Formal analysis:** Paola Benaglio, Jacklyn Newsome, Joshua Chiou, Anthony Aylward, Sebastian Preissl, David U. Gorkin, Kyle J. Gaulton.

**Funding acquisition:** Kyle J. Gaulton.

**Investigation:** Paola Benaglio, Jacklyn Newsome, Sebastian Preissl, David U. Gorkin.

**Methodology:** Paola Benaglio, Jacklyn Newsome, Jee Yun Han, Sierra Corban, Michael Miller, Mei-Lin Okino, Jaspreet Kaur.

**Supervision:** Sebastian Preissl, David U. Gorkin, Kyle J. Gaulton.

**Writing – original draft:** Paola Benaglio, Jacklyn Newsome, Kyle J. Gaulton.

**Writing – review & editing:** Paola Benaglio, Jacklyn Newsome, Sebastian Preissl, David U. Gorkin, Kyle J. Gaulton.

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
