## [Decision Letter · Decision Letter 0]

1 Nov 2022

Dear Dr Gaulton,

Thank you very much for submitting your Research Article entitled 'Mapping genetic effects on cell type-specific chromatin accessibility and annotating complex trait variants using single nucleus ATAC-seq' to PLOS Genetics.

The manuscript was fully evaluated at the editorial level and by independent peer reviewers. The reviewers appreciated the attention to an important topic but identified some concerns that we ask you address in a revised manuscript.

We therefore ask you to modify the manuscript according to the review recommendations. Your revisions should address the specific points made by each reviewer.

Yours sincerely,

Anne O'Donnell-Luria, MD, PhD

Academic Editor

PLOS Genetics

Hua Tang

Section Editor

PLOS Genetics

Reviewer's Responses to Questions

**Comments to the Authors:**

Reviewer #1: Benaglio & Newsome et al investigated non-coding GWAS variant function by generating snATAC-seq on sorted PBMCs from 13 individuals. Using this data, the authors identify chromatin accessibility QTLs (caQTLs), which they linked fine-mapped immune trait variants to putative effector genes using co-accessibility with promoters. The authors provide comprehensive validation of their dataset by demonstrating concordance to previously generated external datasets such as eQTLs and Promoter Capture HiC data in immune cell types as well as bulk caQTL from lymphoblastoid cell lines. The authors functionally validated the allelic effects for a likely causal variant, rs72928038, they provide a sufficiently detailed and thoughtful limitations section in their discussion, and they do not overinterpret their results.

The authors confirm that using snATAC-seq for caQTL mapping is a valuable approach to add to for resolving shared cell type specific differences in cis regulatory elements and provide insight into the mechanism of action for non-coding variant for multiple immune traits. These types of studies have been done in a number human tissues including immune cells, but the current study does represent added value to the field as these approaches mature and sample sizes increase.

Major Issues:

The functional follow up on the T1D rs72928038 variant and its target gene BACH2 is not novel. This functional association was made previously by Robertson et al. (Nature Genetics 2021). While this represents a valuable confirmation, this reduces the novelty of the study. Do this authors have functional validation of another SNP-gene pair they can highlight? Alternatively, can the authors explain how the way they validated this T1D pair is novel and/or value-added compared to the Robertson study?

The uploaded main text figures have a resolution issue that makes several of their axes illegible. In Figure 3C, the trend for enrichment between Cicero predicted co-accessible regions and Promoter Capture HiC interactions appear to increase with distance. How does the distribution of distances between accessible regions and gene promoters in Promoter Capture HiC and Cicero coaccessibility? Are the distributions uniform across genomic distance or could some of the differences between the methods be explained by chromatin conformation/Cicero coaccessibility capturing interactions at different ranges?

Given that sample size is limiting to detect caQTL particularly in rare cell types, it would be useful to discuss how many more donors/cells are necessary for some of the rarer cells (Treg etc) to have comparable power to the more common cell types from this study. It may be useful to run a sampling simulation on a high frequency population (e.g., naive CD4+ T cells) to assess at what point reduced cell numbers hamper caQTL calling power.

While the authors addressed why the colocalization analyses were not performed in the limitations section of the discussion (lines 436-438), this section should be expanded to discuss any expected differences in interpreting the results between the current approach and colocalization.

Minor Issues:

I was confused by the labels of Sup Figure 9 (caQTL population +ASE and caQTL population only & population + ASE). The titles should either be renamed to clarify what is the difference between the groups or elaborated on in the figure legend.

Line 304: (Figure 3f is missing a closing parenthesis

Reviewer #2: In this study by Benaglio et al. the authors perform single nucleus chromatin accessibility sequencing on a collection of PBMCs from 13 individuals. From a total of 96K nuclei profiled they identified 17 immune cell types and subtypes. They then performed allele-specific mapping of caQTLs across the different cell types to identify 6,901 caQTLs at 10% FDR. They further annotate putative target genes using co-accessibility analysis and performed fine-mapping of 16 immune traits. While this is an important study there are some limitations that could be addressed to improve the manuscript.

Specific comments

1. The authors title should provide a more informative and specific title that mentions ‘circulating immune cells’ or PBMCs. Otherwise, it is not clear what cell/tissue was used for the mapping of the caQTLs.

2. In the introduction the authors claim that no other study has performed caQTL mapping in specific cell types using single cell assay. However, there was a recent study published in Nature Genetics by Turner A et al. (PMID 35590109) that mapped cell type specific caQTL using snATAC in coronary artery. The authors should revise the sentence to acknowledge this recent study and cite the paper.

3. The authors obtained an impressive number of cells per sample however there should be more details on how there were more numbers from the “Lab 10X pipeline” when the CellRanger estimates for ATAC often inflates the number based on debris on “cell-like” counts. Also more details on the filtering criteria such as minimum number of fragments per cell and TSS enrichment score cut-off would be useful.

4. The authors provide key cell-specific literature references for the cell type annotation. Have they also considered using an automated immune cell reference such as CellTypist to validate some of these?

5. The number of caQTLs by cell type is high despite by the smaller number of individuals profiled. The mention of 6,901 total caQTLs is based on FDR of 10%, which is quite lenient. The authors should report the number of caQTLs mapped at a more commonly used threshold (5%) in the main text and abstract.

6. The authors should calculate the pi1 statistic (if not already included) to report the number shared true positive between the individual cell type caQTL and the merged bulk caQTL analysis. The authors performed rigorous sensitivity analyses to determine the drivers of the caQTL discoveries, however this additional support would be helpful.

7. The motif enrichment revealed both shared and cell type specific TF motifs. Were the cell type specific enriched motifs correlated with higher differential gene expression for those TF genes? Some further discussion on the redundancy of the TF families as well as cell type specific TFs underlying functionally distinct immune cell types and subtypes for different traits would be helpful.

8. The authors should consider integration of their PBMC snATAC-seq data with PBMC snRNA-seq data to determine whether some of the cell type specific accessible regions correlate with gene expression.

9. In Supp Fig 12, the box plots showing allelic effects on chromatin accessibility at rs9394346 in the bulk sample look statistically significant, compared to the cell type specific box plot for Cyto NK. Also, the box plots for a few other cell types appear to be significant however they have high q-values. The authors should explain whether these peak reads are mapping to a different variant, or this is a limitation of RASQUAL.

10. It might be more helpful to provide cell-type specific bed files in addition to the Supp Table 5.

**Have all data underlying the figures and results presented in the manuscript been provided?**

Reviewer #1: Yes

Reviewer #2: **No: **It appears that data from only 7 of the 13 total individual PBMC samples have been uploaded to GEO.

PLOS authors have the option to publish the peer review history of their article (what does this mean?). If published, this will include your full peer review and any attached files.

Reviewer #1: **Yes: **Andrew D. Wells

Reviewer #2: No

---

## [Decision Letter · Decision Letter 1]

25 Apr 2023

Dear Dr Gaulton,

We are pleased to inform you that your manuscript entitled "Mapping genetic effects on cell type-specific chromatin accessibility and annotating complex immune trait variants using single nucleus ATAC-seq in peripheral blood" has been editorially accepted for publication in PLOS Genetics. Congratulations!

Yours sincerely,

Anne O'Donnell-Luria, MD, PhD

Academic Editor

PLOS Genetics

Hua Tang

Section Editor

PLOS Genetics

Comments from the reviewers (if applicable):

Thank you for your revisions to address the comments from the reviewers.

Reviewer's Responses to Questions

**Comments to the Authors:**

Reviewer #1: The authors have satisfactorily addressed my concerns

Reviewer #2: Thank you for carefully responding to these comments. I have no further questions/concerns. The manuscript is of high quality and should be suitable for publication.

**Have all data underlying the figures and results presented in the manuscript been provided?**

Reviewer #1: Yes

Reviewer #2: Yes

PLOS authors have the option to publish the peer review history of their article (what does this mean?). If published, this will include your full peer review and any attached files.

Reviewer #1: **Yes: **Andrew D. Wells

Reviewer #2: No

**Data Deposition**

http://datadryad.org/submit?journalID=pgenetics&manu=PGENETICS-D-22-00952R1

**Press Queries**

---

## [Editor Report · Acceptance letter]

24 May 2023

PGENETICS-D-22-00952R1 

Mapping genetic effects on cell type-specific chromatin accessibility and annotating complex immune trait variants using single nucleus ATAC-seq in peripheral blood 

Dear Dr Gaulton, 

We are pleased to inform you that your manuscript entitled "Mapping genetic effects on cell type-specific chromatin accessibility and annotating complex immune trait variants using single nucleus ATAC-seq in peripheral blood" has been formally accepted for publication in PLOS Genetics! Your manuscript is now with our production department and you will be notified of the publication date in due course.

With kind regards,

Timea Kemeri-Szekernyes

PLOS Genetics

On behalf of:
